# Effects of Harvest-Aids on Seed Nutrition in Soybean under Midsouth USA Conditions

**DOI:** 10.3390/plants9081007

**Published:** 2020-08-09

**Authors:** Nacer Bellaloui, H. Arnold Bruns, Hamed K. Abbas, Daniel K. Fisher, Alemu Mengistu

**Affiliations:** 1Crop Genetics Research Unit, Agricultural Research Service, USDA, P. O. Box 345, Stoneville, MS 38776, USA; 2Retired Research Scientist, Crop Production Systems Research Unit, Agricultural Research Service, USDA, P. O. Box 350, Stoneville, MS 38776, USA; abruns626@gmail.com; 3Biological Control of Pests Research Unit, Agricultural Research Service, USDA, P. O. Box 67, Stoneville, MS 38776, USA; hamed.abbas@usda.gov; 4Sustainable Water Management Research Unit, Agricultural Research Service, USDA, P. O. Box 127, Stoneville, MS 38776, USA; daniel.fisher@usda.gov; 5Crop Genetics Research Unit, Agricultural Research Service, USDA, Jackson, TN 38301, USA; alemu.mengistu@usda.gov

**Keywords:** harvest aids, seed composition, seed nutrition, defoliant, seed protein, seed sugars, seed oil, soybeans.

## Abstract

Interest in using harvest aids (defoliants or crop desiccants) such as paraquat, carfentrazone-ethyl, glyphosate, and sodium chlorate (NaClO_3_) have become increasingly important to assure harvest efficiency, producer profit, and to maintain seed quality. However, information on the effects of harvest aids on seed nutrition (composition) (protein, oil, fatty acids, sugars, and amino acids) in soybean is very limited. The objective of this research was to investigate the influence of harvest aids on seed protein, oil, fatty acids, sugars, and amino acids in soybean. Our hypothesis was that harvest aid may influence seed nutrition, especially at R6 as at R6 the seeds may still undergo biochemical changes. Field experiments were conducted in 2012 and 2013 under Midsouth USA environmental conditions in which harvest aids were applied at R6 (seed-fill) and R7 (yellow pods) growth stages. Harvest aids applied included an untreated control, 0.28 kg ai ha^−1^ of paraquat, 0.28 kg ai ha^−1^ of paraquat, and 1.015 kg ai ha^−1^ of carfentrazone-ethyl (AIM); 6.72 kg ai ha^−1^ sodium chlorate, 1.015 kg ai ha^−1^ carfentrazone-ethyl; and 2.0 kg ae ha^−1^ glyphosate. Results showed that the application of harvest aids at either R6 or R7 resulted in the alteration of some seed composition such as protein, oil, oleic acid, fructose, and little effects on amino acids. In addition, harvest aids affected seed composition constituents differently depending on year and growth stage. This research demonstrated the possible alteration of some nutrients by harvest aids. This research helps growers and scientists to advance the understanding and management of harvest aids and investigate possible effects of harvest aids on seed nutrition.

## 1. Introduction

The use of the early-maturing soybean cultivars in the Early Soybean Production System (ESPS) has become popular throughout the Midsouth USA [1,2]. Growing early-maturing soybean has allowed producers to avoid risks associated with drought during summer, harvesting delays during the fall and early winter rains, and unfavorable temperatures during flowering and seed pot and seed set [3]. However, it was reported that with the expansion of the use of early-maturing soybean cultivars in the Midsouth (Arkansas, Louisiana, Mississippi, Missouri Bootheel, and West Tennessee), green stems, green pods, or green leaf at harvest increased [4,5]. These conditions can potentially delay the harvest operation and reduce harvest efficiency [6], reduce grade and potential market price, and penalize producers for excessive moisture, foreign materials, splits, and damaged seeds [5,7]. An earlier harvest will allow growers for early delivery and higher market prices [8], and maintain seed quality and composition (seed protein, oil, fatty acids, sugars, and amino acids). Therefore, avoiding all these risks and using harvest aids is critical to ensure that producers maintain profit and seed quality and nutritional value [5].

Previous research on harvest aids reported that the application of glyphosate at five rates, from 0.56 to 3.36 kg, 3 weeks before soybean harvest, resulted in decreased soybean yield [9], and application before physiological maturity resulted in decreased seed weight, emergence, and vigor [10,11]. It was also reported that the intent of pre-harvest, crop harvest aid application was to rapidly dry vegetative and reproductive plant tissues, including seeds, without affecting seed yield and seed quality [12,13,14]. It was reported that the application of glyphosate at 1%, 10%, or 100% of an 890 g ai ha^-1^ rate to soybean near seed maturity had significant effects on germination and/or growth of the resulting F1 generation [15]. Other harvest aids such as carfentrazone and pyraflufen-ethyl can be effective in desiccating broadleaf weeds, although paraquat, used effectively on grass and broadleaf weeds, can result in significant crop injury if applied too early [5]. Although paraquat, carfentrazone, glyphosate, and pyraflufen-ethyl are used as soybean harvest aids, the application timing and rate are critical to avoid yield loss and reduction in seed quality. For example, paraquat applied at R5 (beginning of seed fill) decreased soybean seed quality and yield [16]. It was also reported that paraquat application at R8 (physiological maturity) reduced the number of green stems, pods, and retained green leaves, allowing harvesting 1 to 2 weeks earlier than non-treated soybean [5]. They also found that applying paraquat resulted in less seed moisture, foreign material, and seed damage. Additional research on soybean harvest aids found that applying paraquat at 0.6 and 1.1 kg ha^−1^, glyphosate at 1.7 and 3.4 kg ha^−1^, and ametryn at 1.1 and 2.2 kg ha^−1^, 3 to 4 weeks before harvest, reduced yield compared to when applications were made 2 weeks before harvest [7]. They also found that paraquat was the most effective harvest aid for accelerating soybean harvest and reducing yield and seed weight, but glyphosate was the least effective. Paraquat was also found to be the most effective harvest aid for accelerating soybean harvest [17,18,19,20]. In addition, Boundreaux and Griffin [8], and Blackburn and Boutin [15] found that the application of ametryn and paraquat did not affect seed germination or seedling vigor, while glyphosate applications reduced germination and soybean seedling vigor. They concluded that the application of harvest aid prior to physiological maturity will significantly decrease soybean yields, and glyphosate is not recommended as a harvest aid.

Although information on the effect of harvest aids on seed composition is very limited, available research showed that harvest aids, for example, paraquat, glyphosate, and ametryn, resulted in a decrease of seed oil content when harvest aids were applied 4 and 3 weeks before harvest date [7]. However, when the harvest aids were applied 2 weeks before the harvest date, there were no significant differences in seed oil compared to untreated soybean. The application of ametryn 4 weeks before harvest resulted in lower oil content compared with the other treatments. Protein content increased following the application of ametryn at low and high rates 3 to 4 weeks ahead of harvest [7]. It was concluded that the application of a harvest aid before physiological maturity significantly alters seed composition [7]. Additionally, they added that glyphosate is not recommended for use because of its negative impact on yield and seed quality. If a harvest aid is used to accelerate harvest, harvest aid selection and application timing are critical [7]. Therefore, the objective of this research was aimed at investigating the effects of harvest aids on seed protein, oil, fatty acids, sugars, and amino acids in soybean at different growth stages.

## 2. Results

ANOVA of seed composition constituents showed that, generally, Year, Growth Stage (Stage), and treatments (Treat) were the main factors influencing some seed composition constituents, including seed protein, oil, and some fatty acids, sugars, and amino acids (Table 1 and Table 2).

Based on the *F* values, these three factors are more important than their interactions (Year × Stage × Treat). Therefore, since the *F* value for Year × Stage × Treat interactions was smaller (less significant: its contribution to the model was smaller) compared with that of the effect of Year or Stage or Treat alone, the data were combined across the two years. The most affected compounds were protein, oil, oleic acid, and fructose. Amino acids were affected by Year, Stage, and their interactions: for example, the significant effects of Year on asparagine (ASP), glycine (GLY), valine (VAL), and tryptophan (TRY); the effects of Stage on isoleucine (ISO-LEU) and lycine (LYS); the effects of Year, Year × Stage on threonine (THR), proline (PRO), and serine (SER); and the effects of Stage, Year × Treat × Stage on leucine (LEU) (Table 2). ANOVA for seed composition (protein, oil, fatty acids, sugars, and amino acids) showed that the effect of Year × Stage × Treat interactions were less important than the effects of factors Year, Stage, and Treat alone. Based on F values, seed composition constituents did not change the ranking in two years, indicating that the performance of seed composition constituents had the same pattern across two years. Mean values of seed constituents showed that the application of harvest aids at R6 resulted in the alteration of some seed composition components (Table 3). For example, seed protein increased when paraquat or NaClO_3_ were applied, compared with non-treated plants or when paraquat plus carfentrazone-ethyl or glyphosate were applied, except for carfentrazone-ethyl. The opposite trend for oil was observed in that oil content was higher in all treatments, except for paraquat + carfentrazone-ethyl. No significant effects were observed in stearic acid when soybeans were applied with harvest aids. The application of glyphosate, carfentrazone-ethyl, or paraquat resulted in higher oleic acid compared with the control and when soybean was applied with paraquat plus carfentrazone-ethyl or NaClO_3_. Oleic acid content was no different in the paraquat + carfentrazone-ethyl treatment compared to the non-treated. The application of harvest aids at R6 also showed little changes in linolenic acid, but the change depended on Year and Growth Stage.

The application of harvest aids at R7 resulted in higher protein when glyphosate was applied compared with the control, and lower when NaClO_3_ was applied, but all other harvest aids were similar to the non-treated (Table 3). Oil content was higher than the non-treated when all harvest aids were applied, except for NaClO_3_. The application of paraquat or paraquat plus carfentrazone-ethyl or carfentrazone-ethyl alone to soybean resulted in lower oleic acid. The application of all harvest aids did not affect linolenic, stearic, or palmitic acids at R7.

The application of harvest aids to soybean resulted in minimal influence on sugars, except that harvest aids at R6 showed some changes in fructose (Table 4), as Year, Growth Stage, harvest aid, and their interactions had significant effects on fructose. For example, the application of carfentrazone-ethyl and glyphosate resulted in an increase in fructose, but the application of paraquat or paraquat plus carfentrazone-ethyl resulted in lower fructose (Table 4). The application of harvest aids to soybean had minimal effects on amino acids (Table 5 and Table 6), except for LEU as LEU was significantly influenced by the interactions of Year × Treat × Stage. The significant influence of main effects of Year on amino acids ASP, GLY, ALA, and TRY; Growth Stage on SER, ISO-LEU, and LYS; and the minimal effects of their interactions on THR, SER, and PRO, indicated that the seed content of some amino acids (ASP, GLY, ALA, TRY, SER, ISO-LEU, and LYS) depended on the environmental conditions of each Year and the Growth Stage (Figure 1). 

## 3. Discussion

Information on the effect of harvest aids on seed composition constituents, including seed protein, oil, fatty acids, sugars, and amino acids, is very limited. The increase of protein, resulting from glyphosate application at R7, and the increase of oil, resulted from all harvest aids, except for NaClO_3_, indicated the possible effects of harvest aids on seed composition compounds. The increase in protein at R6, resulting from paraquat, paraquat plus carfentrazone-ethyl, or NaClO_3_ applications; the increase of oil, resulting from the application of paraquat, NaClO_3_, carfentrazone-ethyl, or glyphosate; and the increase of protein and oil due the application of paraquat and NaClO_3_ compared to the non-treated, indicated protein and oil alterations by these harvest aids. The increase of oil when carfentrazone-ethyl or when glyphosate was applied could be due to either the application of these harvest aids or the genetically inverse relationship between protein and oil [21,22,23,24,25]. The increase of oleic acid, due to paraquat, glyphosate, or carfentrazone-ethyl application, indicated the alteration of these acids could be due to translocation or redistribution of fatty acids after harvest aid application at the R6 stage. The higher oleic acid due to glyphosate, carfentrazone-ethyl, and paraquat at R6; and by paraquat, paraquat + carfentrazone-ethyl, and carfentrazone-ethyl at R7 could be due to the effects of harvest aids on desaturases fatty acid enzymes and sugar hydrolysis. The increase of fructose by carfentrazone-ethyl or glyphosate indicated the potential alteration of some harvest aids on specific fractions of sugars.

Although the effects of glyphosate and other herbicides showed conflicting findings on plant growth and seed composition, including mineral nutrients, proteins, oil, and fatty acids [25,26,27], not enough information or recent findings are available to establish the effects of harvest aids on seed composition constituents. The limited previous research available showed that the application of three harvest aids (paraquat, glyphosphate, and ametryn) decreased seed oil content when applied 4 and 3 weeks before harvest date [7]. The application of these harvest aids 2 weeks before the harvest date did not change seed oil composition compared to untreated soybean. On the other hand, when ametryn was applied 4 weeks before harvest, seed oil content was further decreased compared with other treatments. It was also reported that because oil and protein are inversely correlated, protein content increased by the application of harvest aid when applied 4 and 3 weeks before harvest date at high and low rates of application [7]. Only the application of ametryn resulted in higher protein compared with other treatments [7]. Application of harvest aids before physiological maturity significantly alters seed composition, negatively impacting yield and seed quality [7,12,13].

It was also reported that the application of glyphosate at 0.84 kg ha^−1^ at vegetative stages resulted in no yield differences, but resulted in high seed protein, oleic acid, and total amino acid concentrations, with a decrease in linolenic acid concentrations compared with the untreated control [25]. Alterations in these seed composition constituents could be due to the fact that glyphosate inhibits the enzyme 5-enolpyruvyl shikimate-3-phosphate (EPSP) synthase (EC 2.5.1.19), resulting in the reduction of aromatic amino acids and protein synthesis [28], increases of shikimic acid accumulation [29], deregulation of carbon flow into the shikimic acid pathway [30], alteration of soybean seed composition (protein, oil, and fatty acids), and carbon metabolism [23]. The increase of oleic acid and decrease in linolenic acids could be due to a stress response of soybean to glyphosate or indirect effects on fatty acid metabolism and fatty acid desaturases, as suggested by [31], or as a result of carbon metabolism alteration resulting from glyphosate’s effect on desaturases as suggested by others [23].

Evaluation of the effects of glyphosate application on glyphosate-resistant soybeans compared with near-isogenic non-glyphosate parental lines was conducted [27]. The results showed that glyphosate application resulted in significant decreases of nutrient concentrations in shoots and polyunsaturated fatty acid percentages in seeds. They found significant decreases in polyunsaturated linoleic acid (2.3% decrease) and linolenic acid (9.6% decrease) and a significant increase in the monounsaturated fatty acids 17:1n-7 (30.3% increase) and 18:1n-7 (25% increase). They explained that the negative impact of glyphosate was due to the decreased photosynthesis and nutrient availability in glyphosate-treated plants. Others [32] investigated the effect of glyphosate on a soybean glyphosate resistance gene on seed nutrients including Mg, Mn, and Fe, and on yield and amino acids. In a two-year field study in Mississippi, USA, they found no consistent effects of glyphosate on the glyphosate transgene or yield or seed minerals. They found that there were no significant effects on free or protein amino acids in seeds. They concluded that the application of glyphosate appeared to produce random false positives, and that glyphosate or the glyphosate transgene influence the content of minerals measured or seed amino acid composition measured or the yield of glyphosate soybean. Although these results are consistent with previous results [33,34,35], other research reported negative or positive effects on amino acids, seed minerals, and seed composition [25,26,27,36]. Our results showed that the application of harvest aids prior to physiological maturity, such as at R6 or R7, may alter some seed composition constituents, including seed protein, oil, some sugars, and especially fructose. Our results on the effects of harvest aids on seed composition (protein and oil) agreed with those of [7] in that the application of paraquat, glyphosphate, or ametryn resulted in a decrease in seed oil content when applied 4 and 3 weeks before harvest. However, the application of harvest aids 2 weeks before harvest resulted in no significant differences in seed oil compared to untreated soybean. Our results on the effects of harvest aid application on amino acids showed there were alterations in some amino acids, in our case, LEU, agreeing with those of [25,27], where amino acids were affected by increasing glyphosate rates, and their effects were observed whether the application was a single application or were sequential applications at lower rates. The general minimal effects of harvest aids on amino acids, except LEU, in this study, could be due to random false positive or negative effects, agreeing with those reported by others [32]. We must notice here that, since we used different cultivars across two years, the results may be confounded by the cultivar/genotype, although the two cultivars were very close in performance and they belong to the same maturity group to minimize the effect of the cultivar.

## 4. Materials and Methods

### 4.1. Planting and Growth Conditions

An experiment was conducted in 2012 and 2013 on Dundee silty clay (fine-silty, mixed, active, thermic Typic Endoaqualfs) located 2 km north of Elizabeth, MS, on property leased by the USDA-ARS Crop Production Systems Research Unit at Stoneville, MS. Field preparation began for each growing season in the previous autumn by disking the field level and then forming 40 cm high ridges spaced 102 cm apart. Just prior to planting, the ridges were harrowed to form a 40 cm seedbed. The cultivars used in this study were Asgrow AG4303 (Monsanto Corp., St. Louis, MO, USA) in 2012 and Pioneer P94Y23 (DuPont Pioneer Co., Johnston, IA, USA) in 2013. Soybean cultivars were planted in twin rows spaced 25 cm apart and centered 102 cm between rows. Each plot was eight rows in width with each row being 12 m long. Planting dates for the experiment were 23 April 2012 and 18 April 2013, with a planting rate of 30 seed m^−2^ both years. A pre-plant application of 67 kg K ha^−1^ of muriate of potash was applied prior to harrowing, with no other fertilizer applied for the remainder of the season. Weed control was achieved with a pre-emergence application of *S*-metolachlor (Dual Magnum, Syngenta Crop Protection, Greensboro, NC, USA) at 2.15 kg ai ha^−1^, followed by a cultivation prior to the growth stage R1 that also cleared the furrows to facilitate irrigation. Furrow irrigation was applied at a rate of 25.0 mm ha^−1^ on 22 May, 25 June, and 30 July 2012. In 2013, the same rate of irrigation was applied 19 June, 25 June, 1 July, 8 July, 18 July, and 7 August. Irrigation was applied as needed based on the recommendation of the research team.

The harvest aids used were: paraquat (1,1′-Dimethyl-4,4′-bipyridinium dichloride) (Helena Agri-Enterprises, Leland, MS, USA), carfentrazone-ethyl (FMC Corp., Philadelphia, PA, USA), sodium chlorate (NaClO_3_) (Fisher Scientific, Pittsburgh, PA, USA), and glyphosate (*N*-(phosphonomethyl) glycine) (Helena Agri-Enterprises, Leland, MS, USA). Though glyphosate would not be a harvest aid on glyphosate-resistant soybean cultivars as were used in this experiment, it may be used by some growers to kill late-emerging weeds that would interfere with harvest. Treatments applied to sub-plots were: an untreated control, 0.28 kg ai (active ingredient) ha^−1^ of paraquat, 0.28 kg ai ha^−1^ of paraquat + 1.015 kg ai ha^−1^ of carfentrazone-ethyl, 6.72 kg ai ha^−1^ sodium chlorate, 0.015 kg ai ha^−1^ carfentrazone-ethyl, and 2.0 kg ae (acid equivalent) ha^−1^ glyphosate. The application rate of harvest aids was used based on the label recommendations. These treatments were randomized within each whole plot and applied only to the center four rows. Plots were machine-harvested with a Kincaid 8X-P combine equipped with a Juniper HarvestMaster weight system for yield determination approximately 21 days after the final treatment application. Mature seed samples from each sub-plot were collected and used for seed composition analysis.

### 4.2. Seed Protein, Oil, Fatty Acids, and Sugars

Protein, oil, fatty acid, and sugar (glucose, raffinose, and stachyose) contents in mature seeds from each plot were analyzed with a Diode Array Feed Analyzer AD 7200 (Perten, Springfield, IL, USA). Briefly, seeds were ground by a Laboratory Mill 3600 (Perten, Springfield, IL, USA) and approximately 25 g of seed were analyzed for protein, oil, and fatty acid contents according to [24,37,38,39]. Calibration equations were initially developed by the University of Minnesota and upgraded by the Perten company using Perten’s Thermo Galactic Grams PLS IQ software. The calibration equations were established according to AOAC methods [40,41]. Protein, oil, and sugars (glucose, raffinose, and stachyose) were expressed on a dry-matter basis [24,37,38,39,42]. Fatty acids (palmitic, stearic, oleic, linoleic, and linolenic) were expressed on a total-oil basis. Seed glucose and fructose were measured enzymatically using a Glucose (HK) Assay Kit, Product Code GAHK-20 (Sigma-Aldrich Co., St. Louis, MO, USA). The method was previously detailed by Bellaloui et al. [39]. Both glucose and fructose were expressed as mg g^−1^ dry weight.

### 4.3. Seed Amino Acids

Harvested mature seeds from each plot were analyzed for the amino acids alanine (ALN), cysteine (CYS), valine (VAL), methionine (MET), isoleucine (ISO-LEU), leucine (LEU), tyrosine (TYR), phenylalanine (PHE), lysine (LYS), histidine (HIS), arginine (ARG), tryptophan (TRY), asparagine (ASP), threonine (THR), serine (SER), glutamine (GLU), proline (PRO), and glycine (GLY). The analysis was conducted by a near-infrared (NIR) reflectance diode array feed analyzer (Perten, Springfield, IL, USA) as described by [24,25,43,44,45]. Briefly, a sample of approximately 25 g of seed from each plot was ground by a Laboratory Mill 3600 (Perten, Springfield, IL, USA) according to [46,47]. Initial calibration equations were developed by the Department of Agronomy and Plant Genetics, University of Minnesota, St Paul, MN, using Thermo Galactic Grams PLS IQ software developed by the Perten company (Perten, Springfield, IL, USA) and then updated by the Perten company. The quantification of amino acids and calibration equation updating were based on the methods of the Association of Official Analytical Chemists [48] and the use of initial 8540 samples spectra, resulting in accurate estimations of amino acid quantification. Measurement of amino acids content (%) was based on dry-matter.

### 4.4. Experimental Design and Statistical Analysis

The experimental design was a split-plot replicated four times with two harvest aid applications at Growth Stages R6 and R7. The main plot was Growth Stage, and harvest aid treatment was sub-plot. Statistical analyses to evaluate the effect of Year, Growth Stage, and Treatment, and their interactions, were conducted using PROC MIXED (SAS, SAS Institute, 2002–2010) [49]. Replicate Within Year [Rep(Year)], and Stage × Rep (Year) were considered as random effects. Growth stage (Stage) and harvest treatments (Treat) were considered as fixed effects. Mean comparison was conducted by Fisher’s Protected LSD test and the level of significance of *p* ≤ 0.05 was used in SAS (SAS, SAS Institute, 2002–2010) and analyzed according to [50,51].

## 5. Conclusions

The use of harvest aids has become an agricultural practice to improve seed quality and harvest efficiency and allows the harvest of soybean 7 to 10 days earlier when compared to non-treated soybean. However, producers must be careful to not make harvest aid applications too soon, which can result in a reduction in yield [52] and seed nutritional quality. Therefore, to avoid possible negative effects of harvest aids on seed nutrition and composition, especially protein, oil, and oleic acid, management strategies and characterization of factors and mechanisms responsible for the effects of harvest aids need to be understood. Before conclusive recommendations are made, further research is needed. We must notice here that since we used different cultivars across two years, the results may be confounded by cultivar/genotype, although the two cultivars were very close in performance and they belong to the same maturity group to minimize the effect of the cultivar. This information is valuable for the scientific communities and for growers, especially in countries where herbicides are used as harvest aid as a desiccant or for weed control and management.

## Figures and Tables

**Figure 1 plants-09-01007-f001:**
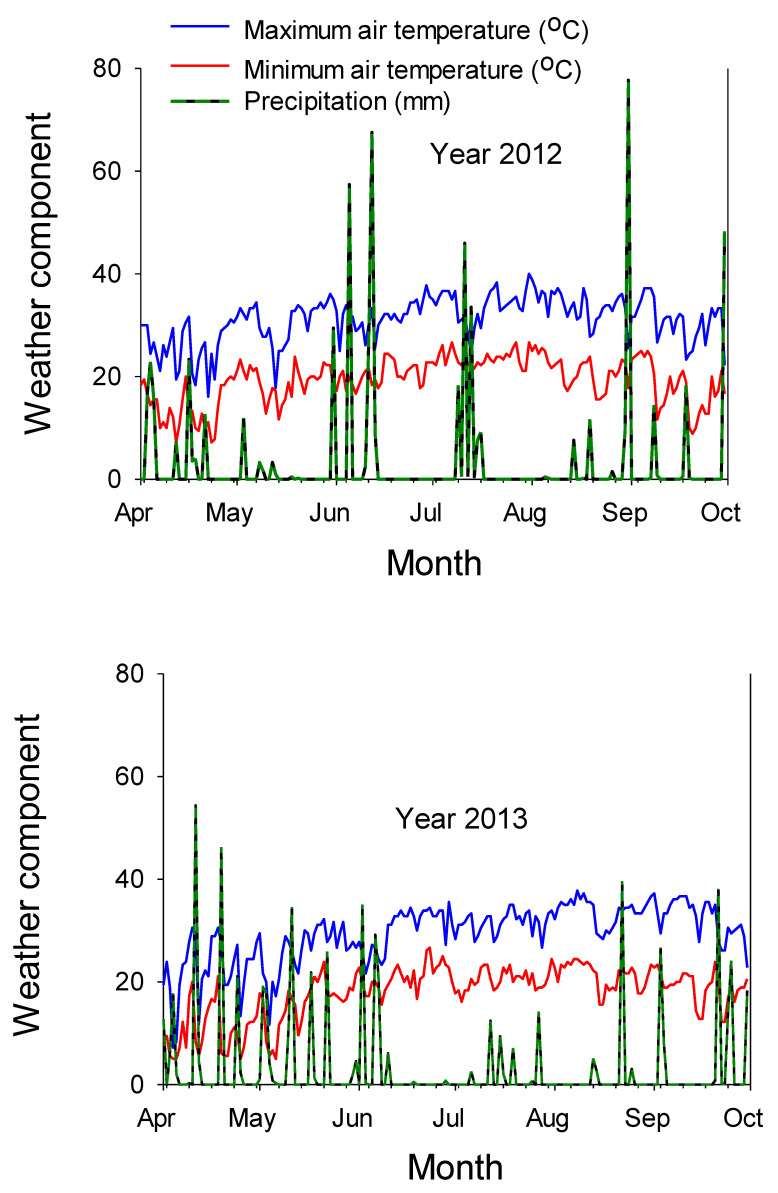
Air temperature (°C) during the growing season in 2012 and 2013. The experiment was conducted in 2012 and 2013 in Stoneville MS. Source: Mississippi State University Extension (2018).

**Table 1 plants-09-01007-t001:** Analysis of variance (*F* and *p* values *^a^*) for the effects of harvest aid application (treatments, Treat: paraquat, paraquat + carfentrazone-ethyl, carfentrazone-ethyl, NaClO_3_, glyphosate) on seed protein, oil, fatty acids (g kg^−1^), and sugars (sucrose, raffinose, stachyose, glucose, and fructose: mg g^−1^). Untreated plants were used as the control. Harvest aids were applied at growth stages (Stage) R6 and R7. The experiment was conducted in 2012 and 2013 in Stoneville, MS.

		**Protein**		**Oil**		**Palmitic**		**Stearic**		**Oleic**		**Linoleic**	
**Effect**	**DF**	***F***	***p***	***F***	***p***	***F***	***p***	***F***	***p***	***F***	***p***	***F***	***p***
Year	1	5.63	*	14.23	***	1.13	ns	44.74	***	0.64	ns	16.55	***
Stage	1	126	***	420	***	6.48	*	1.19	ns	121.88	***	0.95	ns
Treat	5	1.30	ns	28.20	***	0.38	ns	1.18	ns	11.62	***	0.72	ns
Year * Stage	1	1.73	ns	3.59	***	0.08	ns	0.48	ns	0.57	ns	1.96	ns
Year * Treat	5	1.40	ns	7.66	***	0.81	ns	0.33	ns	1.57	ns	0.55	ns
Treat * Stage	5	7.30	***	23.4	***	0.91	ns	1.17	ns	7.9	***	1.18	ns
Year * Treat * Stage	5	0.36	ns	7.5	***	0.44	ns	0.49	ns	0.97	ns	1.08	ns
Residuals		33.7		17.6		6.38		0.95		151		157	
		**Linolenic**		**Sucrose**		**Stachyose**		**Raffinose**		**Glucose**		**Fructose**	
**Effect**	**DF**	***F***	***p***	***F***	***p***	***F***	***p***	***F***	***p***	***F***	***p***	***F***	***p***
Year	1	27.53	***	12.87	**	3.67	ns	3.58	ns	0.73	ns	24.82	***
Stage	1	23.12	***	0.02	ns	0.05	ns	2.25	ns	137	***	144	***
Treat	5	0.95	ns	1.47	ns	1.51	ns	0.55	ns	0.85	ns	22.12	***
Year * Stage	1	15.84	***	0.10	ns	0.95	ns	1.00	ns	0.00	ns	8.78	***
Year * Treat	5	0.99	ns	0.47	ns	1.27	ns	1.38	ns	2.32	ns	11.2	***
Treat * Stage	5	1.35	ns	0.70	ns	0.66	ns	1.08	ns	1.50	ns	4.20	***
Year * Treat * Stage	5	0.91	ns	0.13	ns	0.45	ns	0.75	ns	0.96	ns	1.44	ns
Residuals		35.83		10.09		4.59		0.026		0.065		0.027	

*^a^* * Significance at *p* ≤ 0.05; ** significance at *p* ≤ 0.01; *** significance at *p* ≤ 0.001; DF = degree of freedom; ns = not significant.

**Table 2 plants-09-01007-t002:** Analysis of variance (*F* and *p* values^*a*^) for the effects of harvest aids (treatments, Treat: paraquat, paraquat + carfentrazone-ethyl, carfentrazone-ethyl, NaClO_3_, glyphosate) on seed amino acids asparagine (ASP), threonine (THR), serine (SER), glutamine (GLU), proline (PRO), glycine (GLY), alanine (ALA), cysteine (CYS), valine (VAL), methionine (MET), iso-leucine (ISO-LEU), leucine (LEU), tyrosine (TYR), phenylalanine (PHE), lysine (LYS), histidine (HIS), arginine (ARG), and tryptophan (TRY) (mg g^−1^). Untreated plants were used as the control. Harvest aids were applied at growth stages (Stage) R6 and R7. The experiment was conducted in 2012 and 2013 in Stoneville, MS.

		**ASP**		**THR**		**SER**		**GLU**		**PRO**		**GLY**	
**Effect**	**DF**	***F***	***p***	***F***	***p***	***F***	***p***	***F***	***p***	***F***	***p***	***F***	***p***
Year	1	5.35	*	5.26	**	31.25	***	0.51	ns	10.89	*	31.75	***
Stage	1	0.00	ns	1.87	ns	6.13	**	0.93	ns	1.75	ns	1.21	ns
Treat	5	0.95	ns	0.33	ns	0.59	ns	0.21	ns	1.28	ns	0.46	ns
Year * Stage	1	0.01	ns	3.21	*	3.99	*	0.71	ns	3.24	**	1.78	ns
Year * Treat	5	0.31	ns	1.37	ns	1.11	ns	0.54	ns	1.28	ns	1.68	ns
Treat * Stage	5	0.71	ns	1.88	ns	2.07	ns	0.55	ns	1.74	ns	1.68	ns
Year * Treat * Stage	5	0.97	ns	0.69	ns	0.97	ns	0.26	ns	1.55	ns	0.72	ns
Residual			0.46		0.21		0.51		4.8		0.15		1.32
		**ALA**		**CYS**		**VAL**		**MET**		**ISO-LEU**		**LEU**	
**Effect**	**DF**	***F***	***p***	***F***	***p***	***F***	***p***	***F***	***p***	***F***	***p***	***F***	***p***
Year	1	25.06	***	1.54	ns	15.45	**	8.56	*	0.38	ns	0.23	ns
Stage	1	1.04	ns	0.94	ns	0.43	ns	3.07	ns	2.39	*	5.25	*
Treat	5	0.88	ns	1.90	ns	0.79	ns	1.38	ns	2.49	*	1.79	ns
Year * Stage	1	1.52	ns	1.25	ns	1.66	ns	4.29	ns	0.31	ns	1.94	ns
Year * Treat	5	0.96	ns	0.90	ns	1.76	ns	1.08	ns	2.07	ns	1.49	ns
Treat * Stage	5	2.28	ns	0.85	ns	1.82	ns	1.11	ns	1.19	ns	1.33	ns
Year * Treat * Stage	5	1.29	ns	0.50	ns	1.01	ns	2.81	*	2.46	*	2.72	*
Residual			0.15		0.07		0.34		0.02		0.12		0.56
		**TYR**		**PHE**		**LYS**		**HIS**		**ARG**		**TRY**	
**Effect**	**DF**	***F***	***p***	***F***	***p***	***F***	***p***	***F***	***p***	***F***	***p***	***F***	***p***
Year	1	8.99	**	0.06	ns	0.13	ns	0.12	ns	3.62	ns	7.44	*
Stage	1	0.16	ns	0.07	ns	6.04	**	3.34	ns	0.25	ns	1.95	ns
Treat	5	0.73	ns	1.49	ns	0.09	ns	2.78	ns	1.16	ns	2.23	ns
Year * Stage	1	0.99	ns	0.32	ns	2.73	ns	0.77	ns	0.00	ns	4.65	ns
Year * Treat	5	2.37	ns	0.54	ns	0.74	ns	1.00	ns	0.74	ns	0.48	ns
Treat * Stage	5	1.72	ns	0.28	ns	0.46	ns	1.17	ns	0.45	ns	1.56	ns
Year * Treat * Stage	5	1.18	ns	1.15	ns	0.57	ns	2.27	ns	0.91	ns	1.33	ns
Residual			0.14		0.17		0.23		0.93		0.30		0.006

*^a^* * Significance at *p* ≤ 0.05; ** significance at *p* ≤ 0.01; *** significance at *p* ≤ 0.001; DF = degree of freedom; ns = not significant.

**Table 3 plants-09-01007-t003:** Effects *^a^* of harvest aids (treatments: paraquat, paraquat + carfentrazone-ethyl, carfentrazone-ethyl, NaClO_3_, glyphosate) on seed protein, oil, and fatty acids (g kg^−1^) across two years. Untreated plants were used as the control. Harvest aids were applied at growth stages R6 and R7. The experiment was conducted in 2012 and 2013 in Stoneville, MS.

**Application at R6**						
**Treatment**	**Protein**	**Oil**	**Palmitic**	**Stearic**	**Oleic**	**Linoleic**
Untreated	392	204	101	41.2	243	548
Paraquat	396	208	101	41.4	248	537
Paraquat + carfentrazone-ethyl	394	203	101	41.4	241	542
NaClO_3_	399	208	101	41.9	239	541
Carfentrazone-ethyl	390	221	100	41.2	270	536
Glyphosate	383	230	100	41.5	283	534
LSD	2.42	2.54	0.86	0.39	4.03	3.99
**Application at R7**						
**Treatment**	**Protein**	**Oil**	**Palmitic**	**Stearic**	**Oleic**	**Linoleic**
Untreated	379	228	100	41.6	286	535
Paraquat	379	231	98	40.4	276	535
Paraquat + carfentrazone-ethyl	377	231	100	41.6	272	542
NaClO_3_	376	229	98	41.2	287	533
Carfentrazone-ethyl	379	230	100	40.9	281	542
Glyphosate	384	231	100	41.5	287	537
LSD	2.28	1.56	1.02	0.41	4.50	5.48

*^a^* LSD = Least Significant Difference test, significant at the 5% level. Within each column, the difference between two values is statistically significant if it equals or exceeds the corresponding LSD.

**Table 4 plants-09-01007-t004:** Effects^*a*^ of harvest aids (treatments: paraquat, paraquat + carfentrazone-ethyl, carfentrazone-ethyl, NaClO_3_, glyphosate) on seed linolenic acid (g kg^−1^) and sugars (sucrose, raffinose, stachyose, glucose, and fructose: mg g^−1^) across two years. Untreated plants were used as the control. Harvest aids were applied at growth stages R6 and R7. The experiment was conducted in 2012 and 2013 in Stoneville, MS.

		**Application at R6**				
**Treatment type**	**Linolenic**	**Sucrose**	**Raffinose**	**Stachyose**	**Glucose**	**Fructose**
Untreated	60.3	36.2	4.90	29.2	1.17	0.57
Paraquat	60.8	33.3	4.85	28.2	1.12	0.54
Paraquat + carfentrazone-ethyl	64.7	34.9	4.87	28.5	1.07	0.56
NaClO_3_	56.4	33.6	4.97	26.5	1.11	0.56
Carfentrazone-ethyl	59.3	34.1	4.90	27.8	1.09	0.81
Glyphosate	58.6	33.3	4.90	27.4	1.39	0.77
LSD	4.43	1.28	0.06	0.69	0.07	0.06
		**Application at R7**				
**Treatment type**	**Linolenic**	**Sucrose**	**Raffinose**	**Stachyose**	**Glucose**	**Fructose**
Untreated	54.3	34.5	4.9	28.0	1.64	0.85
Paraquat	54.4	33.8	4.8	28.6	1.79	0.78
Paraquat + carfentrazone-ethyl	53.3	34.3	4.9	27.3	1.84	0.81
NaClO_3_	54.9	32.8	4.8	27.2	1.83	1.05
Carfentrazone-ethyl	56.3	36.3	4.9	28.5	1.76	1.28
Glyphosate	51.6	33.2	4.8	27.5	1.76	1.42
LSD	2.78	1.26	0.06	0.80	0.10	0.103

*^a^* LSD = Least Significant Difference test, significant at the 5% level. Within each column, the difference between two values is statistically significant if it equals or exceeds the corresponding LSD.

**Table 5 plants-09-01007-t005:** Effects^*a*^ of harvest aids (treatments: paraquat, paraquat + carfentrazone-ethyl, carfentrazone-ethyl, NaClO_3_, glyphosate) on seed amino acids asparagine (ASP), threonine (THR), serine (SER), glutamine (GLU), proline (PRO), and glycine (GLY) (mg g^−1^) across two years. Untreated plants were used as the control. Harvest aids were applied at growth stages R6 and R7. The experiment was conducted in 2012 and 2013 in Stoneville, MS.

			**Application at R6**			
**Treatment type**	**ASP**	**THR**	**SER**	**GLU**	**PRO**	**GLY**
Untreated	43.2	14.6	17.4	60.7	19.3	17.5
Paraquat	43.2	14.6	17.5	61.3	19.4	17.6
Paraquat+ carfentrazone-ethyl	43.0	14.5	17.1	61.6	19.2	17.0
NaClO_3_	42.7	14.2	16.6	61.3	19.0	16.7
Carfentrazone-ethyl	43.1	14.4	17.0	61.1	19.2	17.4
Glyphosate	43.3	14.7	17.4	61.1	19.4	18.1
LSD	0.29	0.16	0.26	0.83	0.15	0.44
			**Application at R7**			
**Treatment type**	**ASP**	**THR**	**SER**	**GLU**	**PRO**	**GLY**
Untreated	43.3	14.5	17.5	61.2	19.4	17.7
Paraquat	43.4	14.6	17.6	61.5	19.4	17.5
Paraquat+ carfentrazone-ethyl	42.8	14.6	17.5	60.1	19.4	17.5
NaClO_3_	43.2	14.9	18.0	60.1	19.6	18.3
Carfentrazone-ethyl	42.8	14.6	17.4	60.4	19.1	17.4
Glyphosate	43.1	14.6	17.3	61.1	19.5	17.5
LSD	0.28	0.18	0.32	0.70	0.19	0.49

*^a^* LSD = Least Significant Difference test, significant at the 5% level. Within each column, the difference between two values is statistically significant if it equals or exceeds the corresponding LSD.

**Table 6 plants-09-01007-t006:** Effects^*a*^ of harvest aids (treatments: paraquat, paraquat + carfentrazone-ethyl, carfentrazone-ethyl, NaClO_3_, glyphosate) on the seed amino acids alanine (ALA), cysteine (CYS), valine (VAL), methionine (MET), iso-leucine (ISO-LEU), and leucine (LEU) (mg g^−1^) across two years. Untreated plants were used as the control. Harvest aids were applied at growth stages R6 and R7. The experiment was conducted in 2012 and 2013 in Stoneville, MS.

			**Application at R6**			
**Treatment**	**ALA**	**CYS**	**VAL**	**MET**	**ISO-LEU**	**LEU**
Untreated	17.4	4.3	20.0	5.5	18.7	29.2
Paraquat	17.4	4.2	19.9	5.4	18.7	29.3
Paraquat+carfentrazone-ethyl	17.1	4.3	19.6	5.4	18.8	29.6
NaClO_3_	17.0	4.1	19.5	5.3	19.0	30.2
Carfentrazone-ethyl	17.2	4.3	19.7	5.4	18.7	29.6
Glyphosate	17.4	4.2	20.1	5.4	19.0	29.9
LSD	0.15	0.10	0.20	0.05	0.11	0.26
			**Application at R7**			
**Treatment**	**ALA**	**CYS**	**VAL**	**MET**	**ISO-LEU**	**LEU**
Untreated	17.3	4.2	19.8	5.5	18.6	29.1
Paraquat	17.4	4.2	20.0	5.4	18.9	29.6
Paraquat+carfentrazone-ethyl	17.3	4.2	19.8	5.4	18.5	29.0
NaClO_3_	17.6	3.9	20.3	5.4	18.9	29.6
Carfentrazone-ethyl	17.2	4.3	19.6	5.4	18.5	28.9
Glyphosate	17.3	4.3	19.8	5.5	18.6	29.2
LSD	0.16	0.09	0.23	0.07	0.16	0.32

*^a^* LSD = Least Significant Difference test, significant at the 5% level. Within each column, the difference between two values is statistically significant if it equals or exceeds the corresponding LSD.

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
