# Peer review of "Effects of Harvest-Aids on Seed Nutrition in Soybean under Midsouth USA Conditions"

_plants, 2020, doi:10.3390/plants9081007_

Round 1

Reviewer 1 Report

The reasons for my decision are as follows

The manuscript does not have a hypothesis to demonstrate. It is a descriptive work that its objective is the biochemical characterization of a crop to weed control.

The authors do not report results of herbicidal treatments on crop yield, nor on other agronomic parameters. This information must be provided to infer about the relevance chemical composition.

The results appear confusing because the comparison of means is carried out on all the parameters, but not all are statistically significant.

The herbicides studied cannot be used in all countries and, therefore, the results are not general due to this limitation of the herbicides.

Author Response

Dear editor,

Please receive our revised manuscript entitled “Effects of Harvest-Aids on Seed Nutrition in Soybean Under Midsouth USA Conditions” by Bellaloui et al. 2020

We revised the entire manuscript and made major revision as requested by reviewers and as needed. Our responses to every reviewer are below. We would like to thank the reviewers for their valuable comments. Their comments improved the quality of the manuscript. Please refer to our below responses on one-by-one to the reviewers’ comments. If you have further questions, please let me know.

Thank you for your service,

Dr. Nacer Bellaloui

Research Plant Physiologist

USDA-ARS

Crop Genetics Research Unit

141 Experiment Station Road, P. O. Box 345 Stoneville, MS 38776

Tel:  662.686.5290; Fax: 662.686.5218

Academic Editor:

1-I find it particularly surprising that herbicidal treatments at R6 (seed-fill) did not affect yield. In my opinion, this is a controversial aspect of the study that cannot be dismissed as "data not shown". The authors should show these data, at least as Supplementary material.
2-The fact that a different soybean cultivar was analyzed in each year of the study confounds the effect of the cultivar with that of the year, both in the main effect and in the corresponding interaction effects. This is a poor experimental design that creates non-negligible problems of interpretation, which must be dealt with more carefully in the manuscript.
3-Please, also note that you are submitting for publication in Plants, not in the Canadian Journal of Plant Science.

Our responses:

1-The scientist who collected yield data retired, and there is no way we can get the yield data. This is because we are not able to do any routine office work or lab work due to COVID19. We are allowed to work only in the field these days to accomplish only our critical mission. Therefore, and since our focus in this manuscript is purely on the effects of harvest-aid on seed nutrition, we deleted our statements for yield in the Results section, but we maintained the literature to give a comprehensive report. This way, the manuscript will be devoted only to seed nutrition and not yield, and no misunderstanding will take place. We are currently conducting a second-year field experiment to evaluate harvest-aid Paraquat on yield and seed nutrition in soybean, and we expect the results will be solid as we were lucky to use the same cultivars for two years because the cultivar remained in the market. The results of this research will be available next year and we are planning to submit it to PLANTS. By the way, using different cultivars in two years is not uncommon because seed companies change cultivars over time, and sometimes occur quickly, and the cultivar disappears from the market. We used two cultivars, but both of them were very close for performance and they belong to the same maturity group four (IV). So, using different cultivars should not be a problem; in spite of this, we included in the discussion that soybean genotype may have a confounding effects on the measured variables and we included this in the Discussion section and Conclusions to avoid any misleading understanding, as the editor indicated.

2-We agree with the editor that genotype/cultivar confound other factors, although the cultivars are so close in performance and both belong to maturity group four (IV). We clarified this and explained it further in the Discussion section and Conclusion section for better clarity.

3-Yes, you are right. We already communicated that this manuscript is submitted to PLANTS for the special issue.

Reviewer 1 comment:

Reviewer’s comment

The manuscript does not have a hypothesis to demonstrate. It is a descriptive work that its objective is the biochemical characterization of a crop to weed control.

Our responses: We included a hypothesis and clarified further our objective for better clarity.

Reviewer’s comment

The authors do not report results of herbicidal treatments on crop yield, nor on other agronomic parameters. This information must be provided to infer about the relevance chemical composition.

Our responses: Since the focus of the current manuscript is on harvest-aids application on seed composition/nutrition, we deleted the sections on harvest-aid on yield in the Results section and Discussion section. Also, the scientist who collected yield data retired, and there is no way we can get the yield data. This is because we are not able to do any routine office work or lab work due to COVID19. We are allowed to work only in the field these days to accomplish only our critical mission. We revised the manuscript accordingly as we responded to Editor above. Our ongoing experiment 2019-2020 will be focused on effect of harvest-aid paraquat on yield.

Reviewer’s comment

The results appear confusing because the comparison of means is carried out on all the parameters, but not all are statistically significant.

Our responses: The editor is right. We wanted to present all parameters even if some of parameters are not statistically significant. For better clarity, we included a statement that “LSD value is for each column vertically”. So, each column has its own LSD value separate from other columns or parameters. However, each two means of the same column can be compared, using LSD. Fisher’s Protected LSD test is a common statistical test to compare means and widely used in most prestigious journals such as Crop Science, Frontiers in plant Science, and others, and we used also in our previous articles. We revised all results and reported parameters which are significant for a given treatment, and we also reported others which are not significant for better clarity.

Reviewer’s comment

The herbicides studied cannot be used in all countries and, therefore, the results are not general due to this limitation of the herbicides.

Our responses:

The reviewer is right, and the use of herbicides could be limited compared to USA, Africa, and Latin America. Therefore, although of its limitation use in some countries in Europe, it is still a valuable information to know about the effects of harvest-aids application either if used as desiccant or for weed management. We included a statement in the conclusion that “This information is valuable for the scientific communities and for growers, especially in countries where herbicides are used as harvest-aid as desiccant or for weed control and management.”

Reviewer 2

Only two minor comments for improvement.

It would be better to arrange differently the results, showing first the results of the different treatments and then make the statistical analysis.

Line 186, Supress "A" at the beginning of the sentence.

Our responses: We simplified the tables as suggested by Reviewer 3. I think that the order of tables would work in the current order. Most of the journals follow this format and order, although I have seen some articles they follow as the reviewer suggested. In spite of this, we still value the reviewer’s comment as some few articles and journal present the order the reviewer suggested. We deleted “A” at the beginning of the paragraph.

Reviewer 3

Reviewer’s comments

Comments and Suggestions for Authors

An attempt was made in the study to determine whether harvest-aids affect the chemical composition of soybean seeds. The research problem has been comprehensively investigated. Different chemical agents were applied in the last two growth stages of soybean plants during two growing seasons. This review article summarizes the information available in the existing literature. The manuscript is well written, and it can be very interesting to readers. The review has been carefully prepared. The issues listed below should be addressed during the revision process, although they detract nothing from the scientific merit of the study.

Our responses: We appreciate the reviewer’s positive comment on our study.

Reviewer’s comments

1-In the Results section, please try to keep each table on one page (use single line spacing). Tables 1 and 2 could be simplified by presenting only the significance of the effects exerted by the analyzed factors (leaving only column “P”). Column width should also be adjusted – the tables should have equal column widths (there are blank columns in Tables 3-6). The orientation of the tables should be changed to vertical. Instead of the values of the Least Significant Difference test, it would be better to denote significant differences between groups (treatment and stage) with different superscript letters. Table 6 – not all parameters included in the caption are presented in the table.

Our responses: We would like to keep F value to have an idea about the level of contribution of each parameter even if that parameter is significant. This F value is important, especially for breeders. We will work with the technical team to help us deal with the formatting of text vs. tables and figures and how to make the right orientation and so on. LSD values as presented in tables is the standard way to present it. We added a note in the footnote saying “LSD value is for each column vertically” for better clarity. We fixed the caption of Table 6.

Reviewer’s comments

2-In my opinion, the first two paragraphs in the Discussion section stray from the main topic of the study and should be moved to the Introduction section. Other issues (that are not directly relevant to the subject) can be addressed in the Discussion, but they should mostly serve as an explanation/reinforcement of the Authors’ own findings. The remaining paragraphs are more appropriate for the Discussion section, but several fragments could be moved to the Introduction, in particular those describing previous studies whose results are not directly linked with the current research (e.g. investigating different chemical substances, discussing different effects, etc.).

Our responses: Since the focus of the manuscript now is only harvest-aid effects on seed composition/nutrition, we revised this section as the reviewer suggested, and we kept only what is related to the topic of the manuscript and deleted that section from the Discussion section.

Reviewer’s comments

1-Materials and Methods section:

– why were different soybean cultivars analyzed in each year of the study? The cultivar could have exerted a greater effect on the results than the year of the study;

Our responses: The reviewer is right. We included a statement in the Discussion and Conclusion section that genotype/cultivar has a confounding effect on these parameters. However, we believe that the cultivars were very close in performance and they belong both to the same maturity group, the effect of cultivar on the parameters was minimal. In spite of this, we included a statement to indicate that cultivar may have a confounding effect. Unfortunately, some times companies discontinue cultivars from the market and there is no way to buy it, but rather to replace it with the closet cultivar to the previous one. This is not unusual.

Reviewer’s comments

1– was soybean grown in the same field each year? What was the preceding crop?

2– how (on what basis) was the water demand of seedlings estimated? Was soil moisture content determined in the root zone of plants?

3– how were the doses of the applied chemical substances determined? Were they based on the manufacturers’ recommendations or literature data? In the latter case, the relevant sources should be cited.

Our responses:

Response to the comment 1 and 2: Yes, the experiment was conducted in the same field for two years. The field was watered as needed and based on the recommendation of the irrigation team. Generally, and based on soil type and soil water potential sensors, the field is irrigated when the field does not receive rain 7-10 days water. Our current irrigation research, using water potential sensors in the area where this experiment was conducted, indicated that about -15 kPa represent the water field capacity. At soil potentials of -50 to -60 kPa water stress occurs and irrigation needs to be applied for higher yield. Our research showed that after a regular irrigation (once per 7–10 days), soybean need about 56.8 mm of water every 7 to 10 days to avoid water stress, and this amount could increase to 76.2 mm, depending on the stage of the crop, size of soil cracks, and irrigation water pressure, and other environmental conditions. We added a statement that “irrigation was applied as needed based on the recommendation of the research team.”

3-The doses were applied based on the label. We included a statement indicating that “The application rate of harvest-aids was used based on the label recommendations.”

Reviewer’s comments

The following corrections should also be made:

– phrases such as “In addition, [8,15] found…” should be replaced with “In addition, Boundreaux and Griffin [8], and Blackburn and Boutin [15] found…” –the names of the cited authors should be given, followed by the reference numbers in brackets,

Authors response: Corrected

– fill up page 3,

Authors response: The technical team of Plants will help us with this.

– Table 1 – the first “P” column contains a numerical value instead of a symbol of the significance of differences,

Authors response: Corrected.

– Figure 1 – font size is too large,

Authors response: The technical team will help us

Reviewer’s comments

– remove double spaces in the main text (e.g. line 121),

– Tables 3-6 – arrange treatment groups in the right order,

– line 186 – correct the beginning of the sentence,

– line 309 – decrease indentation in the first line of this paragraph,

– line 326 – insert a space between the numerical value and the unit of measure,

– References section – the numbers of journals should be italicized.

Our responses: All the above comments were addressed as suggested by the reviewer.

Reviewer 4

Comments and Suggestions for Authors

The manuscript entitled: “Effects of Harvest-Aids on Seed Nutrition in Soybean Under Midsouth USA Conditions” aims at investigating the effects of harvest-aids on soybean seeds. In particular, the inspection focused on protein, oil, fatty acids, sugars, and amino acids.

The study is interesting, and it is definitely of interest for the reader of Plants.

The manuscript is clear. The rationale behind the research is properly reported and the state of the art is well described. I personally think the manuscript could be accepted for publication after the correction of a typo at the beginning of Section 3.

At line 186 the authors say: “A The non-effect …” please correct it.  

Our responses:

We appreciate the comments of the reviewer.

We corrected the typo/error as suggested by the reviewer

Reviewer 2 Report

Only two minor comments for improvement.

It would be better to arrange differently the results, showing first the results of the different treatments and then make the statistical analysis.

Line 186, Supress "A" at the beginning of the sentence.

Author Response

(The authors gave the same response as above.)

Reviewer 3 Report

An attempt was made in the study to determine whether harvest-aids affect the chemical composition of soybean seeds. The research problem has been comprehensively investigated. Different chemical agents were applied in the last two growth stages of soybean plants during two growing seasons. This review article summarizes the information available in the existing literature. The manuscript is well written, and it can be very interesting to readers. The review has been carefully prepared. The issues listed below should be addressed during the revision process, although they detract nothing from the scientific merit of the study.

In the Results section, please try to keep each table on one page (use single line spacing). Tables 1 and 2 could be simplified by presenting only the significance of the effects exerted by the analyzed factors (leaving only column “P”). Column width should also be adjusted – the tables should have equal column widths (there are blank columns in Tables 3-6). The orientation of the tables should be changed to vertical. Instead of the values of the Least Significant Difference test, it would be better to denote significant differences between groups (treatment and stage) with different superscript letters. Table 6 – not all parameters included in the caption are presented in the table.

In my opinion, the first two paragraphs in the Discussion section stray from the main topic of the study and should be moved to the Introduction section. Other issues (that are not directly relevant to the subject) can be addressed in the Discussion, but they should mostly serve as an explanation/reinforcement of the Authors’ own findings. The remaining paragraphs are more appropriate for the Discussion section, but several fragments could be moved to the Introduction, in particular those describing previous studies whose results are not directly linked with the current research (e.g. investigating different chemical substances, discussing different effects, etc.).

Materials and Methods section:

– why were different soybean cultivars analyzed in each year of the study? The cultivar could have exerted a greater effect on the results than the year of the study;

– was soybean grown in the same field each year? What was the preceding crop?

– how (on what basis) was the water demand of seedlings estimated? Was soil moisture content determined in the root zone of plants?

– how were the doses of the applied chemical substances determined? Were they based on the manufacturers’ recommendations or literature data? In the latter case, the relevant sources should be cited.

The following corrections should also be made:

– phrases such as “In addition, [8,15] found…” should be replaced with “In addition, Boundreaux and Griffin [8], and Blackburn and Boutin [15] found…” –the names of the cited authors should be given, followed by the reference numbers in brackets,

– fill up page 3,

– Table 1 – the first “P” column contains a numerical value instead of a symbol of the significance of differences,

– Figure 1 – font size is too large,

– remove double spaces in the main text (e.g. line 121),

– Tables 3-6 – arrange treatment groups in the right order,

– line 186 – correct the beginning of the sentence,

– line 309 – decrease indentation in the first line of this paragraph,

– line 326 – insert a space between the numerical value and the unit of measure,

– References section – the numbers of journals should be italicized.

Author Response

(The authors gave the same response as above.)

Reviewer 4 Report

The manuscript entitled: “Effects of Harvest-Aids on Seed Nutrition in Soybean Under Midsouth USA Conditions” aims at investigating the effects of harvest-aids on soybean seeds. In particular, the inspection focused on protein, oil, fatty acids, sugars, and amino acids.

The study is interesting, and it is definitely of interest for the reader of Plants.

The manuscript is clear. The rationale behind the research is properly reported and the state of the art is well described. I personally think the manuscript could be accepted for publication after the correction of a typo at the beginning of Section 3.

At line 186 the authors say: “A The non-effect …” please correct it.  

Author Response

(The authors gave the same response as above.)

Round 2

Reviewer 1 Report

The manuscript can accept